# Distribution and pathogenicity of *Beauveria bassiana* in soil with earthworm action and feeding

Xibei Zhou[1], Wulong Liang[1], Yanfeng Zhang[1], M. James C. Crabbe[1,2,3], Zhumei Ren[1], Yingping Xie[1]*

1 School of Life Science, Shanxi University, Taiyuan, Shanxi, China, 2 Wolfson College, Oxford University, Oxford, United Kingdom, 3 Institute of Biomedical and Environmental Science & Technology, University of Bedfordshire, Luton, United Kingdom

* xieyp@sxu.edu.cn

**Data Availability Statement:** All relevant data are within the paper.

**Funding:** The study was funded by the Shanxi Science [grant numbers 20130311011-3]; Technology Key Project (In Agriculture) of China [grant number 20140311017-6]; and Science and

## Abstract

Earthworm action and feeding have an important impact on a variety of microorganisms in the soil. However, the effects of the earthworm on *Beauveria bassiana*, a common entomopathogenic fungus in the biological control of pests, have been little studied. In this study, the epigeic earthworm species *Eisenia fetida* (Savigny) was selected to evaluate its impact on *B. bassiana* TST05 including its distribution in soil and its pathogenicity to target insects. By testing *B. bassiana* TST05 distribution, biomass in soil, viable spore germination rate, and pathogenicity to insect larvae after passing through the earthworm gut, the results showed that the activity and feeding of *E. fetida* promoted the *B. bassiana* TST05 diffusing downwards in the soil, while decreasing active fungal spores. After passing through the earthworm gut and excretion, the living *B. bassiana* individuals still had activity and pathogenicity to insects. The germination rate of the viable fungal spores was 15.09% and the infection rate to the insect larvae of *Atrijuglans hetaohei* Yang reached 62.35%, 80.95% and 100% after infection at 7 d, 10 d, and 14 d, respectively. The results showed that action and feeding of earthworms promoted the distribution of *B. bassiana* TST05 in soil, but decreased *B. bassiana* viable spores. This study is important for understanding the interaction between earthworms and *B. bassiana* in soil and for guiding the scientific application of *B. bassiana* in the biological control of pests.

## 1. Introduction

Earthworms, belonging to the order Opisthopora and class Oligochaeta in the phylum Annelida, are the soil organisms with the largest biomass in temperate terrestrial ecosystems and play important ecological functions. Earthworms have positive effects on soil aeration, nutrient cycling, soil structure and fertility, plant growth and organic matter accumulation and transformation by their feeding, excretion and movement [1, 2]. In the process of feeding, earthworms devour a large amount of soil, among which microorganisms in the soil are an important food and the source of essential amino acids for earthworms [3]. Earthworm

Technology Innovation Project of Shanxi Education Department [grant number 2019L0065].

**Competing interests:** The authors declare that the research was conducted in the absence of any commercial or financial relationships that could be construed as a potential conflict of interest.

feeding often leads to the reduction of some microorganisms and the proliferation of others [4]. It was reported that earthworms might serve as phoretic hosts to entomopathogenic nematodes and *B. bassiana*. However, it was difficult for entomopathogenic nematodes to survive after they passed through the gut of earthworms, and the transmission of entomopathogenic nematodes by earthworms mainly depended on soil churning and mixing [5–8]. However, there is no report on the important questions, i.e., Whether *B. bassiana* can survive when passing through the gut of earthworms, and whether earthworms can spread *B. bassiana* by excreting earthworm casts.

*Beauveria bassiana* is one of the most common entomopathogenic fungi in the world [9], and has a wide host insect range from multiple orders [10]. In agricultural production, *B. bassiana* has been widely used as a biopesticide in farmlands, orchards, vegetable fields and forests [11]. It has been noted that the number of *B. bassiana* applied into soil can vary greatly with different times, places and treatment methods [12], and soil temperature and humidity are important factors [13–16]. Quintela et al. (1992) suggested that *B. bassiana* in soils with high humidity can be decreased by some competing microorganisms or other inhibiting organisms [14]. However, little attention has been given to the impact of earthworms, one of the largest biomass animals in the soil, on *B. bassiana* [1].

In this study, the epigeic earthworm species *E. fetida* (Savigny) (Opisthopora: Lumbricidae), *B. bassiana* TST05 strain (Moniliaceae: *Beauveria*), and an insect *A. hetaohei* (Lepidoptera: Heliodinidae) were selected to investigate the effects of earthworm action and feeding on the distribution of *B. bassiana* and its pathogenicity to target insects.

The insect *A. hetaohei* is an important pest of walnut fruit in northern China. The mature larvae of *A. hetaohei* coexist with earthworms and *B. bassiana* in the same soil environment for 8–9 months. Under natural conditions, larvae are often infected by entomogenous fungi such as *B. bassiana* in soil. The biological control of *A. hetaohei* using *B. bassiana* and other entomopathogenic fungi has attracted much attention [17]. It was reported that a high infection mortality of mature larvae was achieved using *B. bassiana* strain TST05 to infect *A. hetaohei* which was applied in walnut orchard soil [18]. Therefore, the mature larvae of *A. hetaohei* were selected as the target insects infected by *B. bassiana* strain TST05 as the experimental material in this study.

*Beauveria bassiana* TST05 strain is a highly pathogenic strain that was originally isolated in 2009 by our laboratory from the naturally infected overwintering larvae of *Carposina sasakii* (Matsumura) (Lepidoptera: Carposinidae) in the soil of apple orchards in Xiangfen County, Shanxi Province, China. The strain was identified as *B. bassiana* by molecular technology [19]. The results of sequence alignment were consistent with those of morphological identification. Therefore, the TST05 strain was identified as *B. bassiana*. The biological characteristics of the *B. bassiana* TST05 strain, pathogenicity to host insects, persistence in soil, and compatibility with chemical insecticides were studied in our laboratory [20–22]. At the same time, the strain was deposited in the China General Microbial Species Conservation and Management Center (Beijing, China) under storage number CGMCC4526.

The aim of the study was to learn how to influence the diffusion of *B. bassiana* in the soil by the earthworm action behavior, how to influence quantity and vitality of *B. bassiana* by earthworm feeding and digestion, and whether the viable *B. bassiana* spores in the casts of the earthworms still keep infectivity to the target insects, in order to further understand whether the activities of earthworms in the soil will have an impact on the number, distribution and pathogenicity of *B. bassiana*, and provide a reference for the impact of earthworms on biological control and the application of *B. bassiana* as a biological pesticide.

## 2. Materials and methods

### 2.1 Earthworm, fungus and insects

Earthworm *E. fetida* samples were obtained from the earthworm breeding base in Baiyangdian, Hebei, China, and raised in the animal culture laboratory of the School of Life Science, Shanxi University. Laboratory conditions were 20°C in temperature with 70% relative humidity (RH). The earthworms were maintained in sterilized soil with 40 cm soil thickness and 16% soil moisture. The soil used in the experiment was collected from a forest and was composed of sandy soil 52%, silt 31%, and clay 17%. Fresh fruits and vegetables were put into the soil regularly as a feed for earthworms. Adult clitellate earthworms of similar body size (weight 0.3–0.5 g and length 5.0–5.5 cm) were selected for this study. The earthworms were washed with sterile water for several times until there was no sediment on the surface of the earthworm, then placed in a Petri dish that was covered with wet filter paper and fasted for 48 h to avoid any cross contamination with their casts before they were treated.

The strain used in this test was the isolated and identified strain *B. basiana* TST05 from our previous study [19]. It was inoculated on Potato Dextrose Agar (PDA) medium, and cultivated in the laboratory with 25°C, 75% RH and a photoperiod of 16:8 h (L: D); fungal conidia were collected by 5 d for the study [20].

The mature larvae (in four or five instar stages) of *A. hetaohei* (Lepidoptera: Heliodinidae) were selected as the target insects infected by *B. bassiana* strain TST05. Before the experiment, walnut worm fruit was collected in late August from an experimental walnut orchard in Yuxian County, Shanxi Province, China, and then placed in the laboratory with 50% ~ 70% RH (Longitude 112.58754, latitude 37.79975). When the mature larvae naturally drilled out of the fruit, they were collected and transferred in a culture dish covered with a wet filter for fungal infection study.

### 2.2 Test of the diffusion effect of earthworms on *B. bassiana* spores in soil

The soil column followed the method of Shapiro-Ilan & Brown (2013) [8]. PVC pipes with a diameter of 17cm and a height of 20cm were divided into four equal parts with each part 5cm high, and the four equal parts of PVC pipes were glued together with adhesive tape. The motion range of the earthworm *E. fetida* was confined to the soil column that was stacked with four-section polystyrene plastic tubes. The plastic tubes were fully filled with 7000 g of sterilized soil with moisture content of 16%.

A spore suspension of *B. bassiana* strain TST05 was prepared at $2.5 \times 10^8$ spores/mL, and 25 mL of the suspension sprayed on the surface soil in each column. Fifty adult individuals of *E. fetida* with a biomass of about 20g were placed on the soil surface of each column and kept for four hours at room temperature. Then, the soil columns were maintained at 25°C with 75% RH and a photoperiod of 16:8 h (L: D) for 7 days. The moisture was maintained by adding 12 mL of water every other day. Negative control (CK) groups without earthworms were used for comparison. Three replicates for each treatment and control were performed.

After 7 days, about 1750g soil from each section was transferred to a sterile glass tank and completely mixed by stirring. The soil sample was collected from each section tube and placed into Petri dishes. The soil block was crushed and kept for 15min. Then, 1g dry soil was weighed and diluted 10,000-fold with sterile water.

Then, one mL of the mixture was inoculated on PDA medium and cultured at 25°C and 75% RH with a photoperiod of 16:8 h (L: D). After culturing for 7 days, the fungal colonies were observed and counted referred Xiong's method [19]. Each group was treated for five repeats.

## 2.3 Detection of *B. bassiana* biomass in the earthworm gut and cast

A total of 4 kg dry sterilized soil was inoculated with 640 mL spore suspension ($2.5×10^6$ spores/mL) of *B. bassiana* strain. A total of 250 individuals of *E. fetida* with a biomass of about 100g were transferred into the inoculated soil and maintained at 25°C for 3 days. Negative control (CK) groups were performed without earthworms. After 3 days, in the treatment group, 200 individual earthworms were removed, washed and dissected. The contents of the foregut and midgut of the earthworms were collected. The remaining 50 earthworms were washed, and put into sterile Petri dishes lined with wet filter paper at a shaded place for their casts to be collected. Three replicates were done for both treatments and controls.

The foregut contents, midgut contents and casts of the earthworm *E. fetida* from the treatment group were air-dried in sterile Petri dishes, and the cast granules were crushed into powder by gently pressing them with a sterilized small spoon. One-gram dry samples of the soil as well as the foregut contents, midgut contents and casts were diluted 10,000-fold with sterile water into the mixture. Then, these mixtures were inoculated on PDA medium; the inoculated amount was a 1-mL mixture for each plate. After inoculation, these plate PDA media were cultured at 25°C and 75% RH for 7 days, and their fungal colonies were observed and counted [19]. Each group was repeated for five times.

The soil samples in the control group (CK) without earthworms were collected and the biomass of *B. bassiana* TST05 was detected using the same method as used in the detection of the foregut contents, midgut contents and casts of the earthworm *E. fetida*. The experiment was repeated for five times.

## 2.4 The influence of earthworm digestion on the pathogenicity of *B. Bassiana*

A total of 3kg dry sterilized soil was put into a glass tank (25cm×25cm×40cm), and the thickness of the soil was 20 cm. We inoculated 480 mL of a $5×10^9$ spores/mL spore suspension of *B. bassiana* TST05 into the sterilized soil with the number of *B. bassiana* spores for $3.84×10^{13}$ spores/m$^2$ in a 0-20cm soil layer. It has been reported that the application amount of *B. bassiana* in soil is about $10^{12}$~$10^{14}$ spores/m$^2$ in a 0-20cm soil layer [11, 23, 24]. The number of *B. bassiana* spores in the experiments was consistent with the amount of *B. bassiana* TST05 powder used in field experiments. It has been reported that the biomass of earthworms is 1.60–530.12 g/m$^2$ in a 0-20cm soil layer in farmland [25]. Here, 80 individuals of the earthworm *E. fetida* with a biomass of 33 g were introduced into the inoculated soil, and the biomass of earthworms was 528 g / m$^2$, which was consistent with earthworm biomass in farmland soil. After being transferred into the soil, the earthworms were maintained at 25°C for 3 days. Then, 80 individual earthworms in the treatment group were removed, washed, and put into sterile Petri dishes lined with wet filter paper, and their casts were collected in a shaded space. The experiment was repeated for three times.

In the control group (CK), *B. bassiana* was inoculated but without earthworms, and soil samples were collected after three days. The experiment was repeated for three times. Two grams of the dry cast of *E. fetida* was mixed into 20 mL of sterile water and settled for 10 min. Then, the supernatant was prepared into a spore suspension at a concentration of $1×10^7$ spores/mL. The larvae of *A. hetaohei* were inoculated by soaking in the mixture for 5 s and then transferred in a Petri dish lined with wet filter paper and absorbent cotton at the bottom. Forty larvae in each Petri dish were put in a 25°C incubator to be observed every day for their infection symptoms. There were three replicates for the treatment group. The dead larvae were counted and calculated for their corrected mortality. In the control group (CK), the soil samples were used instead of the earthworm casts. There were three replicates for the control

group. To calculate corrected mortality, the larvae of *A. hetaohei* were soaked in sterile water instead of the mixture. Three replicates were performed.

## 2.5 Statistical analyses

Prior to statistical analysis, all variables expressed as frequencies were arcsine transformed, while quantitative variables were log (x +1) transformed. Data were analyzed using the SPSS 21.0 statistical software package. ANOVA and Tukey's HSD were performed for statistical analyses. Statistical differences were assessed for $P < 0.05$.

## 3. Results

### 3.1 The effect of earthworms on the diffusion of *B. bassiana* in soil

The earthworm *E. fetida* crawled and drilled quickly into the soil once placed on the surface of the soil column (Fig 1A). When the soil columns were taken apart after 7 days, the earthworm distribution in different soil layers was observed, and approximately 80% of earthworms were distributed in the second and third layers of the soil column.

In the treatment group with earthworms (EW), the colonies of *B. bassiana* in the first and second soil layers had the largest numbers (Fig 1B, EW) at 16–21 CFU $g^{-1}$ and 16–19 CFU $g^{-1}$, respectively. Fungal colony numbers were not significantly different between the first and second layers of soil. While, the fungal colonies in the third layer significantly decreased to 10–15 CFU $g^{-1}$. The fourth layer soil contained the least fungal colonies at 1–3 CFU $g^{-1}$, which was significantly different from the other three layers (Fig 1C).

In the control group without earthworms (CK), a large number of *B. bassiana* colonies, 180–260 CFU $g^{-1}$, were isolated from the first layer of the soil column (Fig 1B, CK). However, the second layer soil contained only 0–2 CFU $g^{-1}$ *B. bassiana* colonies, and no fungal colonies were observed in the third and fourth layers of the soil column (Fig 1C). In the control group without earthworms (CK), after 7 days of incubation, the total number of *B. bassiana* spores in the soil column was approximately $3.85 \times 10^9$, but in the treatment group with earthworms, it decreased to $8.4 \times 10^8$. The difference between them was significant.

### 3.2 Effect of earthworm digestion on spore germination of *B. bassiana*

The colony of *B. bassiana* in each sample was mounted by collecting samples from the soil in the control group (CK) and from the foregut contents, midgut contents and casts of the earthworm *E. fetida* in the treatment group; by culturing these samples on plate medium, the colony numbers of *B. bassiana* in each sample were determined. The average numbers of colonies isolated from the soil, foregut, midgut and cast samples were 21.2, 19.4, 11.8 and 3.2, respectively (Fig 2). The soil sample contained the most *B. bassiana*, while the cast sample had the fewest. The *B. bassiana* spores were swallowed with the soil and entered the digestive tract of the earthworm. No significant difference between the *B. bassiana* colonies isolated from the foregut contents and the soil sample, which showed that when they first entered the digestive tract, *B. bassiana* spores were less affected in the foregut, and 91.51% were still alive. The colonies isolated from the midgut contents of the earthworms were fewer in number, indicating that great damage occurred to *B. bassiana* and resulted in a survival rate of 55.66%. After digestion in the midgut of the earthworm, the remaining spores of *B. bassiana* passed through the hindgut and were excreted in the cast where the survival ratio of the spores was only 15.09%.

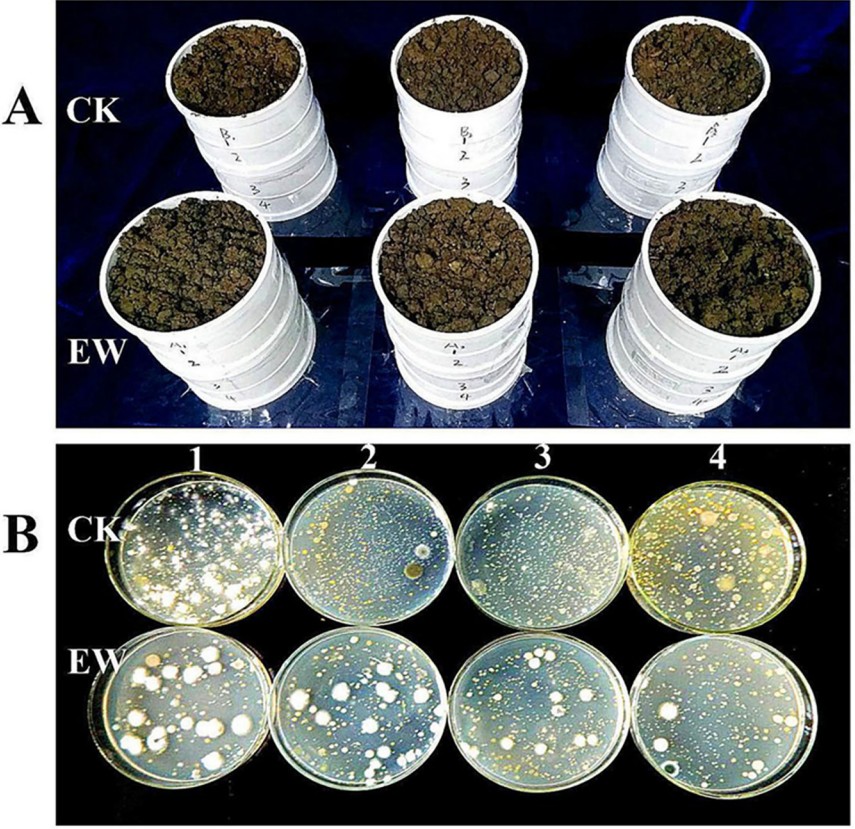

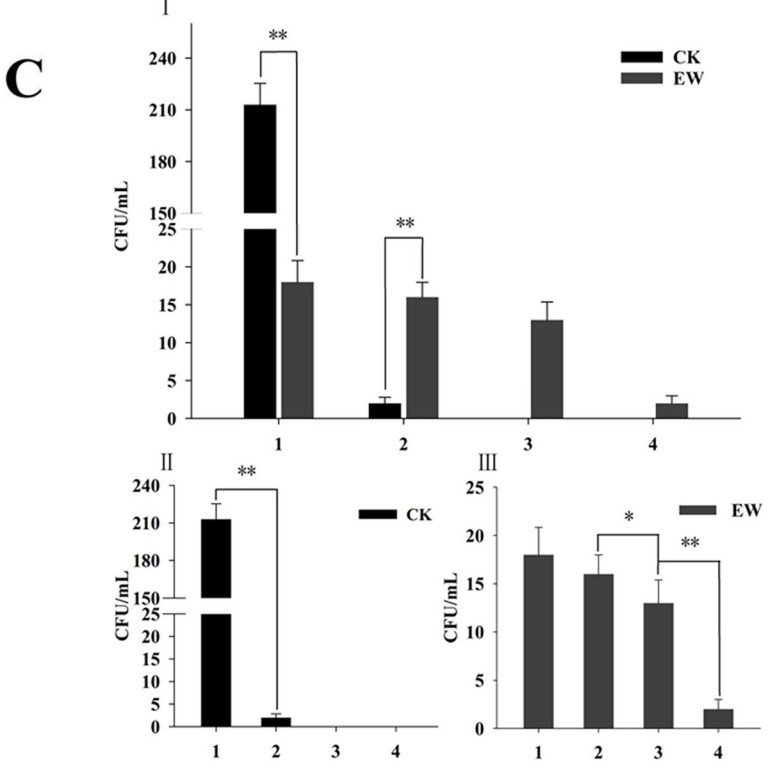

**Fig 1. Effect of earthworm presence on dispersal of *B. bassiana*. (A)**: Soil column on dispersal of *B. bassiana* in the presence of earthworm *E. fetida*; **(B)**: Colony culture of *B. bassiana*; **(C)**: Statistical analyses of the colony number of *B. bassiana* in each layer of soil column. CK: control group with *B. bassiana* only; EW: treatment group included both *B. bassiana* and *E. fetida*. 1–4 represent the first layer (0-5cm), the second layer (6-10cm), the third layer (11-15cm) and the fourth layer (16-20cm) of soil column, respectively.

### 3.3 Pathogenicity of *B. bassiana* in the casts

The infection experiment was conducted on the larvae of *A. hetaohei*, a pest on walnuts, to test the pathogenicity of *B. bassiana* surviving in the casts of *E. fetida*. Either the larvae inoculated with *B. bassiana* in the soil sample (CK group) or these the larvae inoculated with *B. bassiana*

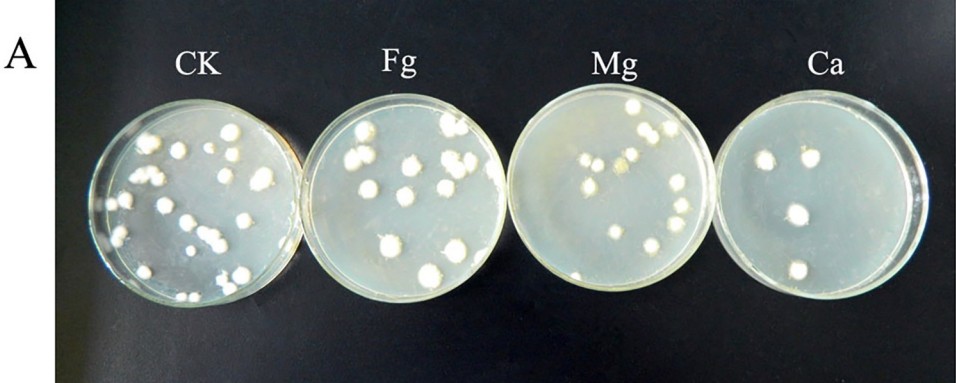

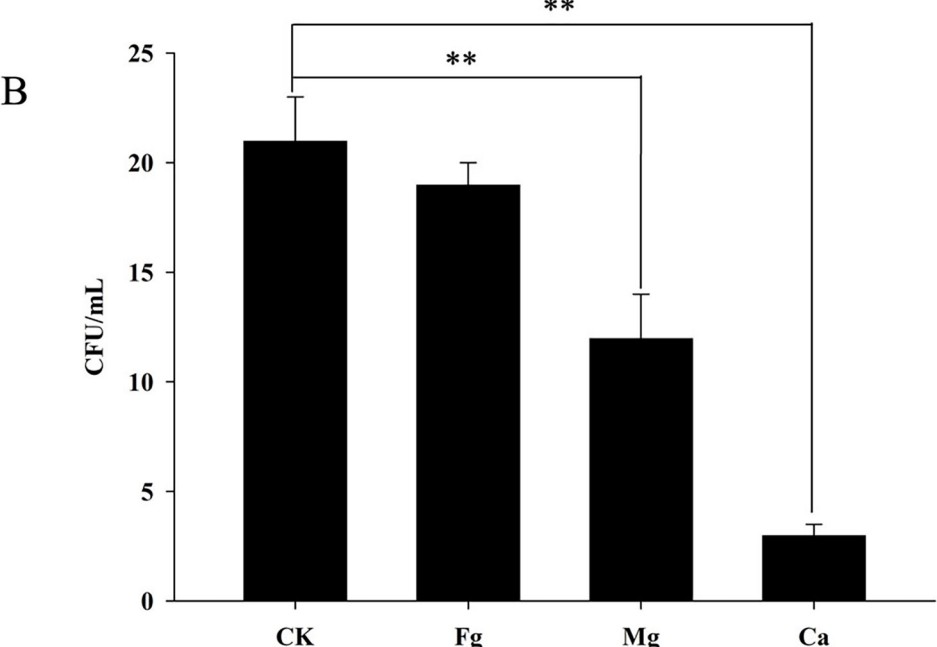

**Fig 2. Effect of earthworm feeding on spore germination of *B. bassiana*. (A)**: The colony culture of *B. bassiana* in the soil, and in the digestive tract and cast of *E. fetida*; **(B)**: Statistical analyses of the colony numbers of *B. bassiana*. CK represents in soil, Fg in the content of earthworm foregut, Mg in the content of earthworm midgut, and Ca in the content of earthworm casts, respectively.

that survived in the cast of *E. fetida* (treatment group) showed the similar infection symptoms, disease courses, and mortality. After infection for two days, the vitality of the larvae decreased. At three days, the body color of the larvae changed to yellow and dark, some larvae began to die, and their bodies gradually became shrunken and shriveled (Fig 3A1 and 3B1). At five days, villous hyphae appeared on the cuticle of the larvae (Fig 3A2 and 3B2). At eight days, the dead larvae became stiff and were partly covered with white hyphae (Fig 3A3, 3B3). At 12 days, the hyphae became thick and completely covered the dead larval bodies (Fig 3A4 and 3B4). At 14 days, dense spores appeared on the body surfaces of the dead larvae (Fig 3A5 and 3B5).

The larval mortality rate at 14 days showed a gradual increase. After three days, the cumulative corrected mortality of the larvae in the treatment group infected by *B. bassiana* that survived in casts of *E. fetida* was 14.61±5.76%, while that in the CK group infected by *B. bassiana* that survived in soil was 15.73±7.64%. At fifth day, the cumulative corrected mortality rates were 34.48±6.03% in the treatment group and 48.28±3.00% in the CK group. At seventh day, the cumulative corrected mortality rates were 62.35±3.68% in the treatment group and 75.29±5.85% in the CK group. At tenth day, the cumulative corrected mortality rates were 80.95±4.17% in the treatment group and 86.90±3.10% in the CK group. At 14 d, the cumulative corrected mortality was 100% in the treatment group and 100% in the CK group (Fig 4). The mortality rate of the larvae in the treatment group was slightly lower than that in the CK group, but the difference was not significant. The median lethal time of the larvae in the treatment group was 6.503 d, while it was 5.583 d in the CK group. Although infection and death of the larvae required more time in the treatment group than in the CK group, the difference between the two groups was not significant (Fig 4). This showed that *B. bassiana* that survived in casts of *E. fetida* still had high infectivity to *A. hetaohei* larvae.

## 4. Discussion

*Beauveria bassiana* is one of the most important entomopathogenic fungi in the world, and the soil is its main habitat [12, 14]. Connections and interactions between *B. bassiana* and other organisms in soil affect the survival, vitality and infectivity of the fungus [10]. Among the numerous soil organisms, earthworms represent the largest animal biomass in soil [1]. It is known that earthworm activities have an important impact on the physiological metabolism and population ecology of some microorganisms in the soil [26–28]. In this study, the earthworm *E. fetida* was used as an example to investigate the effects of earthworm action and feeding on the distribution and pathogenicity of *B. bassiana* TST05 in soil.

Four layer soil columns were employed to assess the effect of *E. fetida* activity on the distribution of *B. bassiana* TST05 in the soil. The results showed that downward diffusion of *B. bassiana* TST05 spores was not obvious in the control group without the presence of the earthworms. The vast majority of *B. bassiana* TST05 spores were still concentrated in the first layer of the soil column. In comparison, the distribution of *B. bassiana* TST05 spores in the soil column changed significantly in the treatment group in the presence of earthworms. *Beauveria bassiana* TST05 colonies were isolated from all four layers of soil samples in the column. This was due to earthworm action to promote *B. bassiana* TST05 spores diffusing downwards, which resulted in the decrease of *B. bassiana* spores in the first layer of the soil column, and the increase in the second, third and fourth layers. The current result is essentially consistent with that of Shapiro-Ilan and Brown (2013) [8], who used the earthworm species *L. terrestris* L., while the earthworm *E. fetida* was used in our study. However, the activities of these two earthworms both promoted the downward diffusion of *B. bassiana* inoculated on the surface of the soil column. In addition, the results also showed that the total number of *B. bassiana* colonies isolated from soil in the treatment group with the presence of earthworms decreased

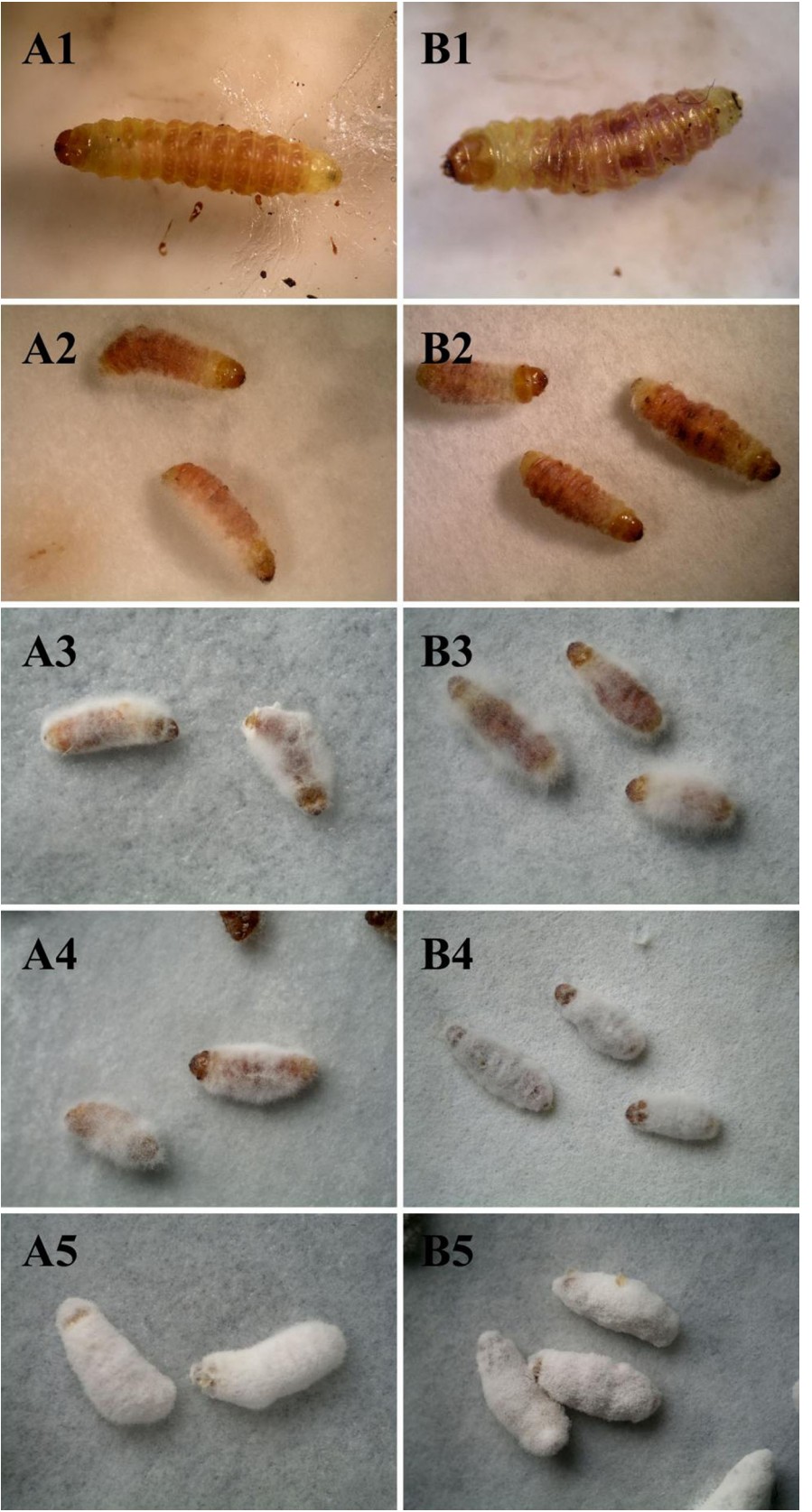

**Fig 3. Disease symptoms of *A. hetaohei* larvae after being inoculated with the *B. bassiana* in the soil (A1~A5) and in the casts of *E. fetida* (B1~B5), respectively.** (A1~A5) show the infection symptoms of larvae being inoculated with *B. bassiana* in soil on the 3rd, 5th, 7th, 10th and 14th days, respectively. (B1~B5) show the infection symptoms of larvae being inoculated with the *B. bassiana* in the casts on the 3rd, 5th, 7th, 10th and 14th days, respectively.

significantly by 78% compared with the control group. It indicated that earthworm activities may promote the downward diffusion of *B. bassiana* in soil but it reduced the number of active spores of *B. bassiana* [10, 29].

The earthworms promote *B. bassiana* diffusion in soil by two methods: one is to attach and carry spores by their body surface, and the second is to feed and excret through their digestive tract. Therefore, it is necessary to study the attachment of *B. bassiana* on the earthworm surface and part of earthworm epidermal mucus.

In the process of earthworm activities, they will devour a large amount of soil so that the fungi in the soil could enter the earthworms' digestive tract, which may be one of the reasons for the decrease in the *B. bassiana* population. It has been found that due to the digestion of earthworms, the composition of fungal populations changed in their gut. The spores of some fungi survive in the midgut environment and begin to germinate and grow more actively in the fresh cast of earthworms. The specific environment of the earthworm gut may comprise the special "filters" and "fermenters" of some soil bacteria and fungi [30]. Moreover, there is no taxonomic relationship in the effects of earthworm digestion on soil bacteria and fungi. Sensitive and resistant populations can be found in the same genus of bacteria or fungi [31]. To observe the effect on the activity of *B. bassiana* after passing through the earthworm gut, *E.*

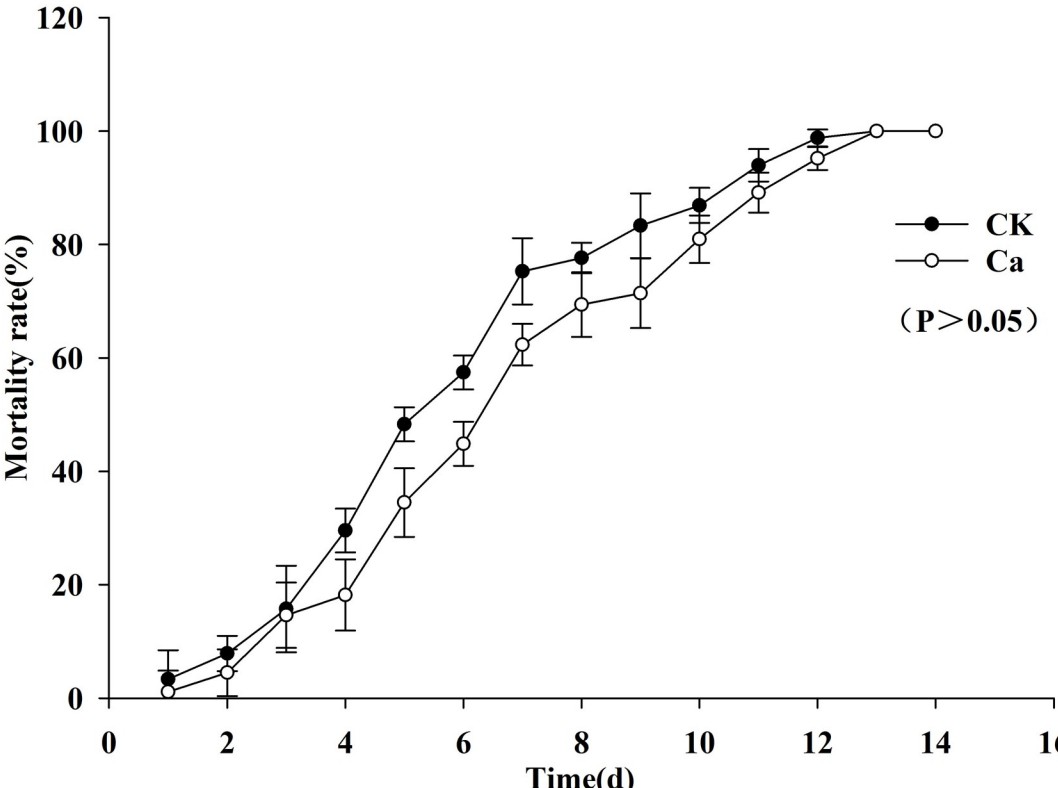

**Fig 4. Corrected mortality of *A. hetaohei* larvae after being inoculated with the *B. bassiana* in the cast of *E. fetida*.** CK: control group of *B. bassiana* in soil; Ca: treatment group of *B. bassiana* in the casts.

*fetida* was fed in soil that was inoculated with *B. bassiana* spores, and then soil samples in the control group (CK) and the foregut contents, midgut contents and cast of *E. fetida* were cultured. Our results have shown that some *B. bassiana* can survive after passing through the digestive tract of earthworms, which can be carried and diffused with earthworm activities and excretion in the soil. However, at the same time, it was found that the total number of active *B. bassiana* spores decreased significantly during this process. Significant changes occurred in the midgut and the casts, and the number of *B. bassiana* spores detected from the midgut content samples was nearly half that of the soil samples; the spores detected in earthworm casts were only 15.09% of the soil samples. This may be due to the digestion of some *B. bassiana* spores by earthworms after passing through the digestive tract of earthworms.

Shapiro-Ilan and Brown (2013) confirmed that *B. bassiana* spores distributed in the soil column still had a high infection rate on *Galleria mellonella* (L.) larvae with the presence of *L. terrestris* L. [8]. In this study, to verify the influence of intestinal digestion of earthworms on the pathogenicity of *B. bassiana* TST05, the walnut fruit bore *A. hetaohei* was selected as the target insect for the infection test. The results showed that the infection processes and symptoms of the two groups were similar, and there were no significant differences in the lethal rate and median lethal time between the two groups. Xiong et al. (2012) used the same concentration of *B. bassiana* TST05 as in the current study to infect the larvae of *Carposina sasakii* Matsumura, and found that the median lethal time was 6.503 days [32], similar to our study. It is inferred that although the number of *B. bassiana* spores is significantly reduced through the earthworm intestinal digestion, the living spores of *B. bassiana* still have strong infectivity to the target insects.

This model takes advantage of the biological and ecological characteristics that *A. hetaohei* larvae fall into the soil after they mature and spend the winter in the soil from September to following May so that the insects are in the same soil environment as earthworms and *B. bassiana*.

## 5. Conclusions

*E. fetida* promoted the diffusion of *B. bassiana* in soil but significantly reduced the total amount of *B. bassiana* in the soil. In the process of earthworm feeding, *B. bassiana* enters the digestive tract together with the soil. After digestion, the surviving spores are excreted with the casts, which is an important way to promote the diffusion of *B. bassiana* and a process of reducing its quantity. After entering the digestive tract of earthworms, the survival rates of *B. bassiana* were approximately 55.66% in the midgut and 15.09% in the casts. However, living *B. bassiana* in the casts still had strong vitality and infection activity, which was manifest in the infection symptoms and mortality rate to the *A. hetaohei* larvae between the treatment and control groups. This is very important for *B. bassiana* in the biological control of pests.

Although this study provides evidence for the effects of earthworm activities, especially feeding and digestion, on the population and infection activity of *B. bassiana* in soil, it is necessary to further study the mechanism of earthworm digestion on *B. bassiana*. The earthworm epidermal mucus is an important immune barrier against microbial invasion. Therefore, it will be important to study whether and how earthworm epidermal mucus inhibits *B. bassiana*.

## Author Contributions

**Data curation:** Xibei Zhou, Wulong Liang, Yanfeng Zhang.

**Writing – original draft:** Xibei Zhou.

**Writing – review & editing:** Wulong Liang, M. James C. Crabbe, Zhumei Ren, Yingping Xie.

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
