## [Decision Letter · Decision Letter 0]

10 Feb 2022

PONE-D-21-30327Distribution and pathogenicity of  Beauveria bassiana  in soil with earthworm action and feedingPLOS ONE

Dear Dr. Xie,

Thank you for submitting your manuscript to PLOS ONE. After careful consideration, we feel that it has merit but does not fully meet PLOS ONE’s publication criteria as it currently stands. Therefore, we invite you to submit a revised version of the manuscript that addresses the points raised during the review process.

We look forward to receiving your revised manuscript.

Kind regards,

Yulin Gao

Academic Editor

PLOS ONE

Dr. Yulin GaoProfessorDepartment of EntomologyInstitute of Plant Protection(IPP)Chinese Academy of Agricultural Sciences(CAAS)2# West Yuan Ming Yuan RoadHaidian District, Beijing, 100193, P.R, ChinaOffice: 01062815930Mobile: 13552643313E-mail: gaoyulin@caas.cn****************************************

Journal Requirements:

Additional Editor Comments (if provided):

I have a great sympathy for non-native English speakers that attempt to publish in English, but these authors will *absolutely* need to have this checked by a native English speaker if they are invited to resubmit and choose to do so.

**Comments to the Author**

Review Comments to the Author

Reviewer #1: This study assessed the impact of the earthworm Eisenia fetida affecting on Beauveria bassiana, including its distribution in soil and pathogenicity to Atrijuglans hetaohei Yang. The result showed that action and feeding of earthworm may promote distribution of B. bassiana TST05 in soil, but decreased the viable spores of B. bassiana. The viable spores of B. bassiana TST05 passed through earthworm gut still had certain germination ability and higher pathogenicity to the insects. This study is important for guiding the scientific application of B. bassiana in biological control of pests. However, several questions should be addressed.

1.Beauveria bassiana TST05 strain that have been identified in the previous studies in line 67-79 do not need to be described in detail in Result section.

2.The statistical difference of fungal colony numbers between layers should been marked in Fig. 1 C and the statistical analyses between the CK group and the treatment group should been labelled in Fig. 5.

3. The discussion section need to be streamlined and logic, e.g. the content same as the result section be deleted and reasonable discussion.

---

## [Author Response · Author response to Decision Letter 0]

10 Mar 2022

Dear Editor, 

Thank you and the reviewers very much for reviewing our manuscript and giving us valuable comments and suggestions. 

According to your advice, we revised all the relevant parts in the manuscript, and responded point by point to the comments as listed below. We highlighted the changes in the revised manuscript by tracking in Word. We also did some edits for the mistakes.

We resubmit here the revised manuscript and hope this revision will be suitable for publication in PLOS ONE.

If you have any questions regarding the manuscript, please feel free to contact me.

With kindest regards, 

Yours Sincerely,

Yingping Xie

Responses to the comments:

Response: The article has been modified according to the format and style requirements of PLOS ONE.

Response: Our colleague Professor James Crabbe has checked the whole MS and re-written the text where appropriate, and so we have invited him as our co-author for his important contribution. 

3. Beauveria bassiana TST05 strain that have been identified in the previous studies in line 67-79 do not need to be described in detail in Result section, L205.

Response: Thanks. Yes, Beauveria bassiana TST05 strain was identified in our previous study, so we deleted the molecular identification process of Beauveria bassiana TST05 strain and supplementary figures S1 and S2 following your suggestion. 

4. The statistical difference of fungal colony numbers between layers should been marked in Fig. 1 C and the statistical analyses between the CK group and the treatment group should been labelled in Fig. 5.

Response: OK, we have finished the statistical analyses and marked them in Figure 1C. We also analyzed the significance between different layers. 

The statistical analysis between the CK group and the treatment group has been labelled in Fig. 5.

5. The discussion section need to be streamlined and logic, e.g. the content same as the result section be deleted and reasonable discussion.

Response: We have simplified and modified the Discussion section, and deleted the same content as in the Results section to make the contents flow smoothly in L285~287, L288~290, L311, L331, L339, L341.

---

## [Decision Letter · Decision Letter 1]

12 Jul 2022

PONE-D-21-30327R1Distribution and pathogenicity of  Beauveria bassiana  in soil with earthworm action and feedingPLOS ONE

Dear Dr. Yingping Xie,

Thank you for submitting your manuscript to PLOS ONE. After careful consideration, we feel that it has merit but does not fully meet PLOS ONE’s publication criteria as it currently stands. Therefore, we invite you to submit a revised version of the manuscript that addresses the points raised during the review process.

We look forward to receiving your revised manuscript.

Kind regards,

Mei Li

Academic Editor

PLOS ONE

Journal Requirements:

Reviewers' comments:

Reviewer's Responses to Questions

**Comments to the Author**

1. If the authors have adequately addressed your comments raised in a previous round of review and you feel that this manuscript is now acceptable for publication, you may indicate that here to bypass the “Comments to the Author” section, enter your conflict of interest statement in the “Confidential to Editor” section, and submit your "Accept" recommendation.

Reviewer #1: (No Response)

Reviewer #2: All comments have been addressed

2. Is the manuscript technically sound, and do the data support the conclusions?

Reviewer #1: Yes

Reviewer #2: Partly

3. Has the statistical analysis been performed appropriately and rigorously? 

Reviewer #1: Yes

Reviewer #2: I Don't Know

4. Have the authors made all data underlying the findings in their manuscript fully available?

Reviewer #1: Yes

Reviewer #2: Yes

5. Is the manuscript presented in an intelligible fashion and written in standard English?

Reviewer #1: Yes

Reviewer #2: No

6. Review Comments to the Author

Reviewer #1: (No Response)

Reviewer #2: Results in this study may be of concern in soil biological interaction because authors provide data on the effects of earthworm action on the distribution of B. bassiana and its pathogenicity to target insects. However, I have got many concerns and there are so many basic errors that I cannot recommend this MS for publication.

CONCERNS:

1. Why the earthworms were maintained in sterilized soil with 40 cm soil thickness? Eisenia fetida is epigeic (0-20cm) and rarely you will find it burrowing in deep soil.

2. The introduction section is poor and do not justify your aims. Line 40-42, why emphasize entomopathogenic nematodes? Line 50, how could the author draw a conclusion that earthworm is a nonhost organism of B. bassiana, is there any evidence? Line 55-57, the sentence is confused. Line 57-62, the life history of insect A. hetaohei should be deleted. Line 64-68, more like within the scope of method. Line 69-77, more like the results of previous study. In this section, please clarify: what gap are you trying to tackle? And why it is important to study the effect of earthworm on distribution and pathogenicity of Beauveria bassianain?

3. The Methods section are not well justified. Line 125, 131, etc., what is the status of sampled soil (dry or wet)? Line 119-120, how about the light condition during culture period? The effective counting concentration is the dilution concentration corresponding to the plate colony count of 20 to 200. Line 126, all samples use the same dilution ratio of 10,000, the CFU of B. bassiana in CK is scientifically inaccurate (Fig.1C). In addition, what is the diameter of the Petri dish? The addition of one milliliter mixture for each plate is excessive.

4. The soil column used in Fig.1a is not described in Method.

5. Line 219-220, is it due to the dilution ratio of 10,000 is too large?

6. Line 155-157, does the midgut material contain intestinal tissue? What components contain in the supernatant? Results in 3.2 showed that great damage occurred in the midgut of the earthworm, and the colonies was conducted based on the midgut contents of the earthworms (without excreting casting). Why collect midgut material after excreted casting? Line 159, Why incubated temperature set as 4°C, far below the ambient temperature for earthworm growth?

7. I found quite difficult to discuss inhibitory effect of the midgut fluid on the spore germination in Fig.3 since the plate is seriously contaminated.

8. The MS requires language editing (really !). Some sentences are awkward. (Line 92; Line 103-104, and many more…)

BASIC ERRORS:

1. Corresponding Author name (see Manuscript Draft in submitted PDF)

2. Latin name (A. hetaohei Yang and A. hetaohei; E. fetida and E. foetida; full name and abbreviation of the Latin name)

3. The use of Arabic numerals. They should never be used at the beginning of sentences. (Line 167)

4. Keywords should not be words found in the title. More, exceed the required number of keywords.

5. Segmentation is confusing (Line 179-191)

6. And many more ...

7. PLOS authors have the option to publish the peer review history of their article (what does this mean?). If published, this will include your full peer review and any attached files.

Reviewer #1: **Yes: **Yin Jiao

Reviewer #2: No

---

## [Author Response · Author response to Decision Letter 1]

29 Aug 2022

Dear Editor, 

Thank you and the reviewers very much for reviewing our manuscript and giving us valuable comments and suggestions. 

We have revised the manuscript and responded point by point to the comments as listed below. We have highlighted the changes in the revised manuscript by tracking in Word. We have also done some edits for the mistakes.

We resubmit here the revised manuscript and hope that this new revision will now be suitable for publication in PLOS ONE.

If you have any questions regarding the manuscript, please feel free to contact me.

With kindest regards, 

Yours Sincerely,

Yingping Xie

Responses to the comments:

1.Why the earthworms were maintained in sterilized soil with 40 cm soil thickness? Eisenia fetida is epigeic (0-20cm) and rarely you will find it burrowing in deep soil.

Response: In this experiment, we used a 40cm thick soil layer to raise earthworms, which aimed to make the soil have better water retention capacity, so that earthworms could survive under stable humidity and temperature.

2.Line 40-42, why emphasize entomopathogenic nematodes?

 Response: Thank you for your question. We have revised this part as “It was reported that earthworms may serve as phoretic hosts to entomopathogenic nematodes and B. bassiana. However, it is difficult for entomopathogenic nematodes to survive when they pass through the gut of earthworms, and the transmission of entomopathogenic nematodes by earthworms mainly depends on soil churning and mixing [5-8]. However, there is no report on the questions: Whether B. bassiana can survive when passing through the gut of earthworms, and whether earthworms can spread B. bassiana by excreting earthworm casts?” in line 36-42.

3.Line 50, how could the author draw a conclusion that a nonhost organism of B. bassiana, is there any evidence?

Response: Thanks for your question, we have revised this sentence as “However, little attention has been given to the impact of earthworms, one of the largest biomass animals in the soil, on B. bassiana [1]”in line 50-52.

4.Line 55-57, the sentence is confused. 

Response: We have changed the description of the sentence as follows: “The insect A. hetaohei is an important pest of walnut fruit in northern China. The mature larvae of A. hetaohei exist in the same soil environment as earthworms and B. bassiana for 8-9 months after entering the soil.” to “The insect A. hetaohei is an important pest of walnut fruit in northern China. The mature larvae of A. hetaohei enter the soils and coexist with earthworms and B. bassiana for 8-9 months.” in line 57-58.

5.Line 57-62, the life history of insect A. hetaohei should be deleted. 

Response: We have deleted Line 58 - 63 on the lifecycle of A. hetaohei. Thanks. 

6.Line 64-68, more like within the scope of method.

Response: Thanks for your suggestions. We have revised this sentence as “It was reported that a high infection mortality of mature larvae was achieved using B. bassiana strain TST05 to infect A. hetaohei which was applied in walnut orchard soil [18].” in Line 61-63.

7.Line 69-77, more like the results of previous study. 

Response: Thank you for your advice. We have revised these sentences as “Beauveria bassiana TST05 strain is a highly pathogenic strain that was originally isolated in 2009 by our laboratory from the naturally infected overwintering larvae of Carposina sasakii (Matsumura) (Lepidoptera: Carposinidae) in the soil of apple orchards in Xiangfen County, Shanxi Province, China. The strain was identified as B. bassiana by molecular technology [19].” in Line 65-68.

8.In this section, please clarify: what gap are you trying to tackle? And why it is important to study the effect of earthworm on distribution and pathogenicity of Beauveria bassiana?

Response: Thank you for your suggestions, we have changed the sentences to “The aim of the study is to learn how to influence the diffusion of B. bassiana in the soil by the earthworm action behavior, how to influence quantity and vitality of B. bassiana by earthworm feeding and digestion, and whether the viable B. bassiana spores in the casts of the earthworms still keep infectivity to the target insects. More importantly, we aimed to understand whether the activities of earthworms in the soil will have an impact on the number, distribution and pathogenicity of B. bassiana, and further provide a basic reference for the impact of earthworms on biological control and the application of B. bassiana as a biological pesticide.” in Line 77-83.

9.The Methods section are not well justified. Line 125, 131, etc., what is the status of sampled soil (dry or wet)?

Response: Thank you for your question, We have added the description of the soil in the manuscript in Line121-123, “The soil sample was obtained from each section tube and placed into Petri dishes. The soil block was crushed and kept for 15min. Then, 1g dry soil was weighed and diluted 10,000-fold with sterile water.”

Line128-129, “A total of 4 kg dry sterilized soil was inoculated with 640 mL spore suspension (2.5×106 spores/mL) of B. bassiana strain.”

10.Line 119-120, how about the light condition during culture period? 

Response: We have added the description of the light condition in the manuscript in Line116-117, “Then, the soil columns were maintained at 25oC with 75% RH and a photoperiod of 16:8 h (L: D) for 7 days.”

11.The effective counting concentration is the dilution concentration corresponding to the plate colony count of 20 to 200. Line 126, all samples use the same dilution ratio of 10,000, the CFU of B. bassiana in CK is scientifically inaccurate (Fig.1C).

Response: Thank you for your suggestions. If different dilution concentrations are used for different experimental samples, there is a concern that the experimental error will also be increased. Therefore, the samples were diluted in the same ratio to reduce the error by repetition. Moreover, the purpose of this test is to compare the changes of spore distribution in the presence of earthworms, but not to accurately understand the specific number of spores. Therefore, the same dilution ratio was used in the test of the diffusion effect of earthworms on B. bassiana spores in soil.

12.In addition, what is the diameter of the Petri dish? The addition of one milliliter mixture for each plate is excessive.

Response: Thank you for your comment. The diameter of the Petri dish we used was 15cm. Adding too much mixture may make it difficult to coat the mixture evenly, but after the comparison of pre-experiments, it was found that adding 1ml of mixture did not have an impact on the experimental results. In order to facilitate observation, statistics and calculation, 1ml of mixture was added in this test, and the experimental effect was feasible.

13.The soil column used in Fig.1a is not described in Method.

Response: Thank you for your advice. We have added the description of the soil column in Line108-112. “PVC pipes with a diameter of 17cm and a height of 20cm were divided into four equal parts with each part 5cm high, and the four equal parts of PVC pipes were glued together with adhesive tape. The motion range of the earthworm E. fetida was confined to the soil column that was stacked with four-section polystyrene plastic tubes. ”

14.Line 219-220, is it due to the dilution ratio of 10,000 is too large?

Response: Thank you for your suggestion. This was pre-tested. When diluted 5000 times, the number of colonies in the first layer of the control group was too large to be counted, and no Beauveria bassiana colonies were observed in the third and fourth layers of the control group. If the dilution ratio of each layer is different, the experimental error might be increased, and the change of colony number cannot be seen intuitively from the picture. Therefore, 10000 times dilution was selected in this experiment. In order to reduce the error caused by dilution, this test was repeated 3 times. In each test, the experiment of the colony numbers was repeated 5 times. According to the test of Shapiro Ilan and Brown, the statistical chart showed that with the presence of Beauveria bassiana but with the absence of earthworms in the soil column for 7 days, the corrected mortality of insects in the soil of third and fourth layers was 0, which was consistent with the conclusion of this experiment. The purpose of this test is to show that with the presence of earthworms, more Beauveria bassiana spores appear in the deeper soil. Even after 10000 times dilution, Beauveria bassiana colonies can be observed in the Petri dishes, while after diluting the soil of the control group in third or forth layer, Beauveria bassiana colonies can not be observed. It was speculated that the number must be less than the group with the presence of the earthworms.

15.Line 155-157, does the midgut material contain intestinal tissue? What components contain in the supernatant? Results in 3.2 showed that great damage occurred in the midgut of the earthworm, and the colonies was conducted based on the midgut contents of the earthworms (without excreting casting). Why collect midgut material after excreted casting? Line 159, Why incubated temperature set as 4°C, far below the ambient temperature for earthworm growth?

Response: Thank you for your suggestions. In this test, the collection of midgut fluid is based on the experiment of Byzov et al.[3]. In the midgut fluid experiment, the supernatant after centrifugation of midgut tissue was used, which may contain a small amount of midgut tissue. At present, there are few reports on the specific components of earthworm midgut fluid, which needs further research and exploration. After earthworm casts, the midguts are cleaner and easier to collect midgut fluid. Intestinal juice is placed at 4ºC in order to keep the midgut fluid fresh. However, due to the reviewer's doubts about Penicillium on the plate, for the precision of this manuscript, all words and pictures related to the midgut fluid experiment have been deleted.

16. I found quite difficult to discuss inhibitory effect of the midgut fluid on the spore germination in Fig.3 since the plate is seriously contaminated.

Response: Thank you for your suggestion. The Penicillium contamination in this test was not caused by the error of experimental operation, but because, in order to collect enough midgut fluid of the earthworms, hundreds of the earthworms were dissected and the midguts were collected in this experiment, and, it is inevitable that there were other miscellaneous microbes in earthworm midguts. Even before using earthworms as experimental materials, raising them in a sterile environment and make them cast cannot remove all miscellaneous microbes in the midguts. Therefore, after mixing the midgut fluid with Beauveria bassiana suspension, Penicillium in the midgut fluid will also grow on the Petri dish and form colonies. After many repetitions of this test, Penicillium colonies were inevitably found in the Petri dish. According to the suggestion of the reviewer, For the precision of this manuscript, all words and pictures related to the midgut fluid experiment have been deleted.

17. The MS requires language editing. Some sentences are awkward. (Line 92; Line 103-104, and many more…)

Response: We have checked all the main text and did some edits. 

 BASIC ERRORS:

1.Corresponding Author name (see Manuscript Draft in submitted PDF)

 Response: Have Corrected the Corresponding Author name

2.Latin name (A. hetaohei Yang and A. hetaohei; E. fetida and E. foetida; full name and abbreviation of the Latin name)

Response: Have Corrected Latin name 

3.The use of Arabic numerals. They should never be used at the beginning of sentences. (Line 167)

Response: Have corrected in line 151.

4.Keywords should not be words found in the title. More, exceed the required number of keywords.

Response: The keyword is modified to "Eisenia fetida, cast, TST05, pathogenicity, spore germination, Infection, Insect, Entomopathogenic fungi".

5.Segmentation is confusing (Line 179-191) 

Response: Have corrected in Line 164-175.

---

## [Decision Letter · Decision Letter 2]

26 Sep 2022

Distribution and pathogenicity of  Beauveria bassiana  in soil with earthworm action and feeding

PONE-D-21-30327R2

Dear Dr. Xie,

We’re pleased to inform you that your manuscript has been judged scientifically suitable for publication and will be formally accepted for publication once it meets all outstanding technical requirements.

Kind regards,

Mei Li

Academic Editor

PLOS ONE

Additional Editor Comments (optional):

Reviewers' comments:

Reviewer's Responses to Questions

**Comments to the Author**

1. If the authors have adequately addressed your comments raised in a previous round of review and you feel that this manuscript is now acceptable for publication, you may indicate that here to bypass the “Comments to the Author” section, enter your conflict of interest statement in the “Confidential to Editor” section, and submit your "Accept" recommendation.

Reviewer #1: All comments have been addressed

Reviewer #2: All comments have been addressed

2. Is the manuscript technically sound, and do the data support the conclusions?

Reviewer #1: Yes

Reviewer #2: Yes

3. Has the statistical analysis been performed appropriately and rigorously? 

Reviewer #1: Yes

Reviewer #2: Yes

4. Have the authors made all data underlying the findings in their manuscript fully available?

Reviewer #1: Yes

Reviewer #2: Yes

5. Is the manuscript presented in an intelligible fashion and written in standard English?

Reviewer #1: Yes

Reviewer #2: Yes

6. Review Comments to the Author

Reviewer #1: The manuscript has been significantly improved throughout the review process, so i think the revised manuscript is suitable for publication in Plos One.

Reviewer #2: (No Response)

7. PLOS authors have the option to publish the peer review history of their article (what does this mean?). If published, this will include your full peer review and any attached files.

Reviewer #1: **Yes: **Jiao Yin

Reviewer #2: No

---

## [Editor Report · Acceptance letter]

3 Oct 2022

PONE-D-21-30327R2 

Distribution and pathogenicity of *Beauveria bassiana* in soil with earthworm action and feeding 

Dear Dr. Xie:

I'm pleased to inform you that your manuscript has been deemed suitable for publication in PLOS ONE. Congratulations! Your manuscript is now with our production department. 

Kind regards, 

on behalf of

Dr Mei Li 

Academic Editor

PLOS ONE